# 1,5-Diarylidene-4-Piperidones as Promising Antifungal Candidates Against *Cryptococcus neoformans*

**DOI:** 10.3390/antibiotics14090883

**Published:** 2025-09-01

**Authors:** Elise Courvoisier-Dezord, Hugo Ragusa, Axelle Grandé, Louise Denudt, Yolande Charmasson, Frédéric Dumur, Didier Siri, Marc Maresca, Malek Nechab

**Affiliations:** 1Aix Marseille Univ, CNRS, Centrale Marseille, iSm2, 13013 Marseille, France; elise.courvoisier-dezord@univ-amu.fr (E.C.-D.); axellegrande@gmail.com (A.G.); louisedenudt@gmail.com (L.D.); yolande.charmasson@univ-amu.fr (Y.C.); frederic.dumur@univ-amu.fr (F.D.); 2Aix Marseille Univ, CNRS, ICR UMR 7273, F-13397 Marseille, France; hugoragusa13@gmail.com (H.R.); didier.siri@univ-amu.fr (D.S.)

**Keywords:** antifungal, *Cryptococcus neoformans*, piperidones, curcumins

## Abstract

Background/Objectives: The present study investigates the antifungal potential of 1,5-diarylidene-4-piperidones. Methods: These compounds were synthesized via Claisen–Schmidt condensation, and their antifungal efficacy was tested against *Cryptococcus neoformans*, a yeast recently qualified as a critical priority pathogen by the World Health Organization, through determination of their minimum inhibitory concentration (MIC). We designed and synthesized a series of piperidones to explore structure–activity relationships. Results: Systematic modification of the substituent pattern revealed that tetrabutoxy groups exhibited potent activity (MIC of 7.8 µM), surpassing standard antifungals like fluconazole. The selectivity index (SI) values confirmed their safety profile across various human cells. Docking analysis demonstrated that these compounds target sterol 14-demethylase, suggesting potential inhibition of ergosterol synthesis as a mechanism of action. Interestingly, the compounds also demonstrated broad-spectrum activity against other pathogenic yeasts and fungi, including *Candida* and *Aspergillus* species, and against fluconazole-resistant strains. Conclusions: These findings underscore the potential of 1,5-diarylidene-4-piperidones as promising antifungal candidates with a favorable safety profile.

## 1. Introduction

The infectious agent *Cryptococcus neoformans* is a capsulated yeast present in soils and organic debris that is transmitted by air. Invasive fungal infection with *C. neoformans* is a serious cause of morbidity in HIV/AIDS immunocompromised patients. The most common clinical forms of infection are septicemia, meningoencephalitis, and pneumonia [1]. In HIV/AIDS immunocompromised patients, infection by *C. neoformans* is life-threatening, mainly due to meningoencephalitis, accounting for approximately 10–15% of AIDS-associated deaths worldwide, with 1,000,000 new cases and over 600,000 deaths each year [2,3]. Regarding immunocompetent patients, although infection by *C. neoformans* is frequent, with up to 70% of healthy children having serum antibodies against *C. neoformans* confirming exposure, fatal outcome is less frequent, the infection being either naturally cleared or persisting as a latent and asymptomatic form [4,5,6]. *C. neoformans* was recently identified as a top-priority fungal pathogen, and even as a “critical priority” by the World Health Organization (WHO) due to its high mortality and morbidity and its resistance to treatment [7,8,9]. Indeed, the mortality rates due to *C. neoformans* infection have been reported to be as high as 41–61% and 8–20% for HIV-positive and HIV-negative patients, respectively, despite antifungal therapy [10]. The standard treatment is amphotericin B combined with flucytosine [11]. However, both agents are toxic, and laboratory monitoring of these patients is needed. Moreover, intravenous administration of amphotericin B is required, limiting, therefore, the access of patients from resource-constrained countries [12,13]. In infections with no meningoencephalitis, fluconazole has been a primary antifungal agent for managing cryptococcosis; however, excessive use of azole drugs has led to drug resistance. Indeed, the ability of *Cryptococcus* to alter its genomic architecture in the face of antifungal stress primarily occurs through a phenomenon referred to as heteroresistance. Bongomin et al. [14] reported in 2018 that the resistance to fluconazole in clinical strains of *C. neoformans* was present in about 18% of the 4995 clinical isolates described in 29 studies from 1988 to 2017 included in the EMBASE and MEDLINE databases. More recently, an analysis of heteroresistance conducted in Brazil on clinical strains of *C. neoformans* revealed that 85% of the isolates showed a moderate level of resistance to fluconazole (≥16 µg/mL) and 40% a high level (≥32 µg/mL), underlying the importance of this phenomenon and its increase over time [15]. The toxicity of the used molecules, their poor ability to cross the blood–brain barrier, and their ability to cause the development of resistance explain why mortality and morbidity caused by *C. neoformans* remain high despite actual antifungal therapy, necessitating the search and identification of novel antifungal drugs directed toward *C. neoformans*.

In this context, curcumin-derived 1,5-diarylidene-4-piperidones have attracted attention for their broad biological activities, including anti-cancer [16], anti-bacterial, anti-inflammatory, and antifungal [17,18,19]. Their structural features make them promising candidates for overcoming current therapeutic limitations. The α,β-unsaturated carbonyl moiety can act as a Michael acceptor, particularly to thiol nucleophiles, enabling specific interactions with fungal targets [20,21,22]. This assumption prompted us to design a new piperidone library to investigate their potential antifungal activities. The synthesis of a curcumin-derived drug is straightforward and based on a Claisen–Schmidt reaction, in which a ketone reacts with two molecules of benzaldehyde analogs in the presence of sodium hydroxide to reach benzylidene structures. Motivated by the simplicity and the potential of such molecules, we aimed to target a series of benzylidene compounds, given the urgent need for the development of new antifungal treatments in the face of drug resistance. We focused on structural modifications of piperidone compounds, including aryl and amine substitutions, to investigate their structure–activity relationships (SARs) and identify potent antifungal candidates against C. *neoformans* and other clinically relevant fungi.

## 2. Results

### 2.1. Synthesis of Piperidones

To target the structure–activity relationship (SAR) of novel curcumin analogs, preparation of several 1,5-diarylidene-4-piperidones (Figure 1) was planned by varying their electronic and/or hydrophobic properties (Figure 1). The Claisen–Schmidt condensation of piperidones and aldehyde derivatives yielded twenty-six compounds in a one-step procedure without the need for column chromatography, as precipitation in ethanol is conclusive to obtaining pure products with yields ranging from 48 to 86% [23,24]. We introduced structural diversity through the modification of alkoxy groups, N-alkyl functions, and lateral moieties. Except for **3h** and **3iso-h** (Figure 1), all new compounds were isolated through recrystallization from ethanol as intensely colored solids. The structural identity and substitution patterns were confirmed by the ^1^H NMR and ^13^C NMR techniques. Mass spectroscopy confirmed the exact masses, while ^1^H NMR showed a purity of >95%. While the general procedure is reported in the Materials and Methods, the structural analyses of the new piperidones are described in the Appendix A.

### 2.2. Antifungal Activities and Toxicity of Synthesized Piperidones

The antifungal activities of the twenty-three piperidone compounds were first determined against *C. neoformans* using the minimum inhibitory concentration (MIC) assay and are reported in Table 1. The antifungal potency was found to be modest for the monosubstituted aryl series, **1a**–**1f**, as well as for compound **2j**, with MICs ranging from 125 to 250 μM, regardless of the substitution (R^1^ = OMe, Me, CN, NO_2_, Cl, F, CF_3_). Disubstituted methoxy and allyloxy groups (**2a** and **2g**) afforded even worse results. However, dibutoxy compound **2h** exhibited a significant ability to inhibit the growth of *C. neoformans* (MIC of 31.2 μM). Compound **2i**, derived from piperonal, showed weak activity with an MIC of 250 μM. Piperidone **2l**, possessing a quinoline scaffold analogous to those reported by Shingate [25], was tested but showed lower efficiency (MIC = 250 μM). Since butoxy groups appeared to exhibit the best activity against *C. neoformans*, we decided to synthesize and evaluate compound **2m**, which contains six butoxy groups, thereby enhancing the hydrophobic ability of the piperidone. Compound **2m** displayed a good MIC of 62.5 µM, but it was less efficient than **2h**. Bicyclic piperidone **2n** was found to be inactive against *C. neoformans*. We then varied the nature of N-alkyl substitution. The results indicated that the cyclopropyl, Boc, mesyl, and Cbz groups were detrimental to the activity. Benzyl amines (e.g., butenafine) are known to be antifungal agents [26] with activity on the cell membrane through squalene epoxidase binding [27]. We decided, therefore, to synthesize compound **3h**, which significantly improved activity, presenting an MIC as low as 7.8 μM. To confirm this effect, we compared the activity of **2k** and **3j** by switching from ethyl alkyl to the benzyl group on the nitrogen atom. Again, the results showed that benzyl substitution (compound **3j**) was beneficial, presenting a lower MIC (62.5 μM) compared to the ethyl-substituted piperidone, **2k** (MIC = 250 μM). Testing compound **2o**, which contains one butoxy group on each aromatic moiety, again proved detrimental, with an MIC of 125 μM (vs. 7.8 μM for **3h**), highlighting the importance of the presence of two butoxy groups on each aromatic moiety. A similar trend was observed when the position of the alkoxy group was changed from ortho to meta (**3iso-h** being less active than **3h**, with MICs of 62.5 and 7.8 µM, respectively). The comparison between compound **3a** (MIC > 250 µM), with methoxy instead of butoxy groups, and compound **3h** strongly suggests that the hydrophobic nature and the length of the alkyl chains are crucial for the activity of these piperidone derivatives. The longer butyl chains in **3h** likely facilitate stronger hydrophobic interactions with a specific region (hydrophobic pocket) on the target protein, leading to enhanced biological activity. In addition, to underscore the significance of the piperidone core, we synthesized and evaluated 1,5-diarylidene cyclohexanone analogs **8**–**10h**, which demonstrated no activity with MIC > 250 µM. These results suggest that the amphiphilic character of piperidones—due to the presence of four alkyl chains and a nitrogen atom—plays an important role. In addition to the hydrophobic interactions identified in our docking study, the benzyl group in **3h** may enhance antifungal potency through a mechanism similar to that of known benzylamine antifungals such as butenafine. Indeed, benzylamines are reported to interact with fungal membranes and enzymes such as squalene epoxidase, leading to potent antifungal effects [26,27]. In **3h**, the benzyl substitution may, therefore, provide a dual advantage. Interestingly, a control test revealed that compound **3h** was more active than fluconazole (MIC of 25 µM), an antifungal recommended to treat *C. neoformans* infection [28], particularly in low-income and middle-income countries [29].

To better evaluate the therapeutic potential of the active compounds, their toxicity was then measured using various human cells, i.e., A498 (kidney), BEAS-2B (lung), Caco-2 (intestine), HaCaT (skin), and HepG2 (liver) cells (Figure 2). For comparison, the toxicity of fluconazole was also determined (Figure 3). The CC_50_ values, i.e., the concentrations causing a 50% reduction in the cell viability, were determined from graphs and are provided in Table 1.

The CC_50_ values of the compounds range from 18.3 to >1000 µM, depending on the compound and the cell model tested. Compounds with monosubstituted aryl groups in para-positions **1b**, **1c**, **1e**, **1f**, and **2j** showed the highest toxicity (CC_50_ of 18.3 to 313.9 µM). Introducing a substituted alkoxy group, like in **1a** and **2o**, was beneficial, providing lower toxicity (CC_50_ of 64.9 to >1000 µM). In addition to higher antifungal activity, the disubstituted butoxy groups (**2h**) revealed low toxicity (CC_50_ of 140 to 696.6 µM). When the aryl groups were connected to acetal, a detrimental effect was observed on the toxicity of compound **2l** (CC_50_ of 30.0 to 89.5 µM). However, incorporating a third butoxy group in the **2m** drug afforded an even lower toxicity (CC_50_ of 663 to >1000 µM). We then evaluated the effect of the substitution on the nitrogen atom by introducing alkyl groups. Benzyl group **3h** revealed a small elevation in toxicity when it was compared to **2h** (CC_50_ of 42.4 to 628.6 µM), whereas compound **4h**, with the cyclopropyl group, showed the lowest toxicity regarding all the cell lines tested (>1000 µM).

For comparison, fluconazole provided a CC_50_ ranging from 841.2 to >1000 µM (Figure 3 and Table 1). In order to further identify the most promising candidates, the selectivity indexes (SIs) of each compound (i.e., the ratios of CC_50_ to MIC values) were calculated (Table 1).

These investigations revealed that **2h**, **2m**, **3iso-h**, and **3h** have the best safety/activity profile with very interesting SIs of up to 80.5. Importantly, **3h** gave the highest SI values, ranging from 5.4 to 80.5, depending on the human cell model considered. Although less active than **3h** (with MIC values on *C. neoformans* of 7.8 and 62.5 µM, respectively), **2m** was found to be the least toxic of the active compounds, with CC_50_ values ranging from 663.7 to >1000 µM and SIs ranging from 10.6 to >16. For comparison, fluconazole yielded SIs ranging from 33.6 to >40, close to the ones obtained with the safest compounds of this study.

The spectrum of antifungal activity of the more active compounds (i.e., **2h**, **2m**, **3iso-h**, and **3h**) was further screened using various fungal strains (yeasts and filamentous species) infecting humans or plants (Table 2 and Table 3). Regarding human pathogenic yeasts (Table 2), **2h**, **2m**, **3iso-h**, and **3h** showed MIC values similar to the ones obtained on *C. neoformans* when tested against *Candida auris* and *C. glabrata* (MIC of 7.8 to 31.2 µM) but were found to be less active against *C. albicans* and *C. tropicalis* (MIC of 62.5 to 250 µM). Regarding filamentous fungi (Table 3), for the human pathogen *Aspergillus fumigatus* and the plant pathogen *A. flavus*, **3h** was the only active compound (MIC of 31.2 µM), with **2h**, **2m**, and **3iso-h** being found to be less active or not active (MIC of 125 to >250 µM). **2h**, **2m**, **3iso-h**, and **3h** were also found to be less active or not active against the plant pathogens *Colletotrichum graminicola*, *Fusarium graminearum,* and *Penicillium verrucosum* (MIC of 125 to >250 µM), except for **3h**, yielding an MIC of 62.5 µM on *P. verrucosum*. Regarding the plant pathogens *Magnaporthe oryzae* and *Microdochium bolleyi* and the human pathogen *Trichophyton rubrum*, **2h**, **2m**, **3iso-h**, and **3h** displayed good activities (MIC of 7.8 to 62.5 µM), except for **2m** on *M. bolleyi* (MIC > 250 µM) and **3iso-h** on *T. rubrum* (MIC of 125 µM), with **3h** yielding the lowest MIC on those strains (i.e., 7.8 µM). Interestingly, it must be noted that yeasts and filamentous fungi found to be resistant to fluconazole (MIC of 250 to >1000 µM) were still sensitive to **2h**, **2m**, **3iso-h**, and **3h**. Thus, once again, **3h** showed the strongest antifungal activity, with the lowest MIC values across all strains.

Importantly, the twenty-three piperidone compounds were tested against Gram-positive and Gram-negative bacteria belonging to the ESKAPE group and were found to be inactive with MICs > 250 µM for all compounds, except **2k**, yielding an MIC of 250 µM on the Gram-negative bacteria *A. baumannii* and *E cloacae*, and **2l** and **3j**, yielding an MIC of 250 µM on the Gram-positive bacterium *S. aureus*. This demonstrates that the compounds possess antimicrobial activity selectively directed against yeasts and fungi that can be explained by the specific targeting of fungal enzyme(s).

### 2.3. Molecular Modeling

It has been demonstrated that similar compounds are able to interact and inhibit cytochrome P450 14α-sterol 14-demethylase (CYP51), an enzyme involved in ergosterol biosynthesis in fungi [25]. Docking analysis was, therefore, performed to evaluate the affinity of the best piperidone molecules with the same sterol 14-demethylase (the one from *T. cruzi*) using its crystal structure (PDB code: 3KHM). Except for piperidone **2m**, which contains three butoxy groups, the results indicated that compounds **2h**, **3h**, and **3iso-h** effectively interacted within the active site of the CYP51 complex. The docking binding energy for **3h** was found to be −8.2 kcal/mol, similar to that of fluconazole (binding energy = −8.1 kcal/mol), which supports the fact that **3h** may act through the inhibition of ergosterol synthesis, as previously shown for other piperidone compounds described in the literature [22,25]. In this molecular docking study, all docked compounds (**2m**, **2h**, **3h**, and **3iso-h**) formed interactions with Hem300 and hydrophobic interactions with numerous amino acid residues (Table 4 and Figure 4, Figure 5, Figure 6, Figure 7, Figure 8 and Figure 9). These hydrophobic interactions with butoxy chains may explain the difference in activity between **3h** (MIC = 7.8 µM) and its methyl analog, **3a** (MIC > 250 µM). In addition, piperidones **2h** and **3h** showed T-shaped interactions with Tyr103, whereas **3iso-h** exhibited T-shaped interactions with His294 and Phe290.

In order to study the stability of the ligands in the active sites, we performed Molecular Dynamics (MD) calculations on the best positions of **fluconazole, 2h**, **2m**, **3h**, and **3iso-h** in CYP51. We calculated the Root Mean Square Deviation (RMSD) of the ligands with respect to starting geometries during the MD trajectories. The results are presented in Figure 10 and Table 5. The MD results show that all complexes are stable, with low RMSD mean values (less than 0.3 nm). **2m** seems to be the least stable, as suggested by the docking results.

## 3. Discussion

*C. neoformans* is a significant human pathogen, but other fungi should also be considered when developing antifungal drugs. *Candida* spp. infect mainly immunocompromised patients but also immunocompetent ones, leading to more than 1.5 million cases of bloodstream infection or invasive candidiasis and around 900,000 deaths every year [30]. In addition to *C. albicans*, among the various *Candida* spp. that infect humans, *C. auris*, *C. glabrata*, and *C. tropicalis* have attracted more attention due to their increasing prevalence and their ability to develop resistance to antifungal drugs used in medicine, including fluconazole [31,32,33,34,35]. Similarly, *Aspergillus fumigatus* infects mainly immunocompromised patients and is estimated to cause over 600,000 deaths annually [36]. Although they do not infect humans, fungi that infect plants and crops are also important to consider when developing antifungals, as they cause massive losses (for example, every year, *Fusarium graminearum* causes losses of around 28 million metric tons of wheat grain valued at USD 5.6 billion [37] and/or produces mycotoxins harmful for humans and animals) [38].

This study demonstrates that 1,5-diarylidene-4-piperidones represent a promising scaffold for antifungal drug development, particularly against Cryptococcus neoformans. The structure–activity relationship analysis (Figure 11) revealed that substitution with butoxy groups significantly enhanced antifungal potency, as observed for compounds **2h** and **3h**, with the latter showing MIC values as low as 7.8 µM. The comparison between methoxy and butoxy analogs highlights the critical role of hydrophobic chain length in driving activity, likely through enhanced interactions within hydrophobic pockets of the target enzyme.

When testing the spectrum of antifungal activity in the best piperidone compounds, the data showed that in addition to possessing good activity against *C. neoformans*, **2h**, **2m**, **3iso-h**, and **3h** also demonstrated antifungal activity against significant human and plant pathogens, such as *C. albicans*, *C. auris*, *C. glabrata*, *A. fumigatus*, and *T. rubrum*, as well as *A. flavus*, *M. oryzae*, *M. bolleyi*, and *P. verrucosum*. In all cases, **3h** was identified as the most active piperidone compound, with MIC ranging from 7.8 to 62.5 µM. Docking analysis and Molecular Dynamics (MD) calculations suggest that **2h**, **2m**, **3h**, and **3iso-h,** as with fluconazole, interact with CYP51, suggesting that these molecules act through the inhibition of this enzyme, altering the synthesis of ergosterol. Although docking analysis and MD calculation strongly indicate that the mechanism of action of piperidone involves CYP51 inhibition, future studies are required to confirm this hypothesis through enzyme inhibition assays and/or ergosterol level measurements. Importantly, fungi resistant to fluconazole were found to still be sensitive to **2h**, **2m**, **3h**, and/or **3iso-h**. This observation is not in contradiction with the docking data, indicating that these molecules target ergosterol biosynthesis like fluconazole. Indeed, the resistance to fluconazole is not due to a mutation in its target but typically involves a reduction in its uptake and/or an increase in its efflux in resistant strains [39]. The fact that fluconazole-resistant strains remain sensitive to **2h**, **2m**, **3h**, and/or **3iso-h** is highly significant due to the growing incidence of fluconazole resistance in fungal strains infecting humans, including *Candida* species and *A. fumigatus* [40,41].

## 4. Materials and Methods

### 4.1. Chemistry

All reagents and solvents were purchased from Sigma Aldrich (Saint-Quentin-Fallavier, France), FluoroChem (Penrose Dock, Ireland) or TCI (Zwijndrecht, Bengium) and used as received without further purification. Mass spectroscopy was performed by the Spectropole of Aix-Marseille University. ESI mass spectral analyses were recorded with a 3200 QTRAP (Applied Biosystems SCIEX, Nottingham, UK) mass spectrometer. The HRMS mass spectral analysis was performed with a QStar Elite (Applied Biosystems SCIEX, Framingham, MA, USA) mass spectrometer. Elemental analyses were recorded with a Thermo Finnigan EA 1112 (Thermo Fisher Scientific, Waltham, MA, USA) elemental analysis apparatus driven by the Eager 300 software. ^1^H and ^13^C NMR spectra were determined at room temperature in 5 mm o.d. tubes on a Bruker Avance 400 spectrometer or a Bruker Avance 300 spectrometer (Bruker, Billerica, MA, USA) at the Spectropole: ^1^H (400 MHz), ^1^H (300 MHz), ^13^C (100 MHz), and ^13^C (75 MHz). The ^1^H chemical shifts were referenced to their solvent peak, CDCl_3_ (7.26 ppm) or DMSO_d6 (2.50 ppm), and the ^13^C chemical shifts were referenced to their solvent peak, CDCl_3_ (77.16 ppm) or DMSO_d6 (39.52 ppm). All compounds were prepared at analytical purity up to accepted standards for new organic compounds (>95%), which was checked by high-field NMR analysis.

N-alkylpiperidin-4-one (10 mmol) was dissolved in 20 mL of ethanol and benzaldehyde derivative (20 mmol), and 40% NaOH (2 equiv.) was then added at 0 °C. The solution was stirred at room temperature overnight. The yellow precipitate was filtered off, washed with cold ethanol, and dried under vacuum (48–86% yield). Spectral data, structure, and NMR spectra of the compounds are provided in the Appendix A.

### 4.2. Antimicrobial Activity

The antifungal effect of the compounds was measured following the reference methods for yeasts and molds as previously described [42]. Reference yeast strains tested were *Candida albicans* (DSM 10697), *C. auris* (DSM 21092), *C. glabrata* (DSM 11226), *C. tropicalis* (DSM 9419), and *Cryptococcus neoformans* (DSM11959). Filamentous fungi tested were either human pathogens (i.e., *A. fumigatus* (DSM 819) and *Trichophyton rubrum* (DSM16111)) or plant pathogens (i.e., *Aspergillus flavus* (DSM 1959), *Colletotrichum graminicola* (DSM 63127), *Fusarium graminearum* (DSM 1095), *Magnaporthe oryzae* (gift from Richard O’Connell, UMR Bioger, Paris Saclay) [43], *Microdochium bolleyi* (DSM 62073), and *Penicillium verrucosum* (DSM 12639)). Yeast suspensions were prepared by resuspending colonies collected from PDA plates in sterile NaCl 0.9% solution. Yeasts were then diluted to 1–2 × 10^3^ yeasts/mL in RMPI media supplemented with glucose (1.8%) buffered with MOPS (final concentration of 0.165 mol/L (pH 7.0)). For filamentous fungi, conidia were collected from mycelium grown on PDA plates using a sterile solution of 0.9% NaCl supplemented with Tween at 0.1%. After counting under a microscope, dilutions at 2–3 × 10^4^ conidia/mL were also prepared in MOPS-buffered RMPI media supplemented with glucose. Diluted yeast or fungi were then exposed to increasing concentrations of compounds (1/2 serial dilution) in 96-well plates, with pure DMSO (maximal concentration of 1%) used as a negative control. Plates were incubated at 35 °C for 24–48 h for human-infectious yeasts and filamentous fungi (i.e., *A. fumigatus* and *T. rubrum*) or 25 °C for 48–72 h for the other filamentous fungi from the environment or plant-infectious fungi. Minimum Inhibiting Concentrations (MICs) were determined as the lowest concentrations of compounds that totally inhibited the growth of the fungi. The antibacterial effect of the compounds was measured through liquid MIC determination using Gram-negative (*Acinetobacter baumannii* (DSM 30007), *Enterobacter cloacae* (DSM 30054), *Klebsiella pneumoniae* (DSM 26371), and *Pseudomonas aeruginosa* (ATCC 9027)) and Gram-positive (*Enterococcus faecalis* (DSM 2570), *Enterococcus faecium* (DSM 20477), and *Staphylococcus aureus* (ATCC 6538)) bacteria from the ESKAPE group following the National Committee of Clinical Laboratory Standards procedure, as previously described [44]. In all cases, MICs were determined using both visual evaluation and optical density measurements (at 600 nm). Experiments were conducted at *n* = 2–3.

### 4.3. Cytotoxicity Studies

The toxicity of the compounds was tested using human cells (i.e., A498 (human kidney cell line), BEAS-2B (normal human airway epithelial cells), Caco-2 (human intestinal cell line), HaCaT (normal human skin cells), and HepG2 (human liver cell line)) as previously described [45]. Cells were obtained from ATCC (Molsheim Cedex France) except HaCaT cells that were obtained from (Creative Bioarray, Shirley, NY 11967, USA). Cells were maintained at 37 °C in a 5% CO_2_ incubator in Dulbecco’s modified essential medium (DMEM) supplemented with 10% fetal calf serum (FCS), 1% L-glutamine, and 1% antibiotics (all from Thermo Fisher Scientific, Illkirch–Graffenstaden, France). Cells grown in 75 cm^2^ flasks were detached using trypsin–EDTA solution (from Thermo Fisher), counted using Mallasez’s chamber, and seeded into 96-well cell culture plates (Greiner bio-one from Dominique Dutscher, Brumath, France) at approximately 10^4^ cells per well. After 24–48 h, when the cells reached 80–90% confluence, wells were aspirated, and increasing concentrations of compounds (from 0 to 1000 µM, ½ dilution) were added to the cells, with DMSO (maximal concentration of 1%) used as a negative control. After 48 h of incubation at 37 °C in a 5% CO_2_ incubator, wells were aspirated, and the cell viability was measured by adding 100 µL of resazurin solution at 0.03 mg/mL to phosphate-buffered saline with calcium and magnesium chloride (PBS^++^). After 1 h of incubation at 37 °C, the fluorescence intensity (Ex 530 nm/Em 590 nm) of the wells was measured using a microplate reader (Biotek, Synergy Mx, Colmar, France). The fluorescence values were normalized by the negative controls (DMSO-treated cells) and expressed as the percentage of cell viability. The cytotoxic concentration 50 (i.e., CC_50_) values of compounds corresponding to the concentrations that caused a reduction of 50% in cell viability were calculated using the GraphPad^®^ Prism 8 software. Experiments were performed at *n* = 3.

### 4.4. Molecular Docking

All the ligands were fully optimized with the Gaussian16 Rev. A03 package [46] at the HF/6-31G(d) level of theory. The atomic charges were computed at the HF/6-31G(d) level of theory with the RESP scheme. For docking studies, the X-ray crystal structure of sterol 14alpha-demethylase (CYP51) from Trypanosoma cruzi in complex with inhibitor fluconazole was obtained from the Protein Data Bank (PDB) (PDB code: 3KHM). The enzyme was prepared by removing fluconazole and then adding hydrogen atoms and atomic charges with the AutoDock Tools 1.5.7 software. The molecular docking studies were carried out with the AutoDock Vina 1.2.5 software [47] with default parameters. The docking box was set at 60 × 60 × 60 Å and was centered at x = 2.270 Å, y = −23.537 Å, and z = 16.924 Å (center of fluconazole). The position drawings and interaction calculations were created with Protein*Plus* and BIOVIA Discovery Studio. The 3D active sites were drawn with PyMOL 3.1.4.1. Details of the molecular docking analysis are provided in the Appendix A.

### 4.5. Molecular Dynamics

Molecular Dynamics (MD) calculations were performed with the Gromacs 2021 [48] package. The complexes were solvated in a quasi-cubic box containing ~16,000 water molecules. Chloride counterions were added to obtain a neutral system. Concerning the force fields used in the MD simulations, we used the GAFF [49] force field and TIP3P [50] model to describe the water molecules. In order to optimize the simulation box, we performed an NPT calculation at 300 K and 1 bar over 400 ps with a time step of 0.5 fs. After this first stage, we performed an NVT trajectory at 300 K over 100 ns with a time step of 0.5 fs. We kept the last 99.5 ns of the trajectory for data analysis calculations.

## 5. Conclusions

This study highlights the potential of 1,5-diarylidene-4-piperidones as effective antifungal agents, particularly against *Cryptococcus neoformans* and other clinically significant fungi. Among the synthesized compounds, **3h** emerged as the most promising candidate, displaying superior antifungal potency. Its ability to inhibit fluconazole-resistant strains further underscores its therapeutic value. Docking studies suggest that these compounds act by targeting sterol 14α-demethylase, disrupting ergosterol biosynthesis in fungal cells. Additionally, their broad-spectrum antifungal activity and low cytotoxicity support their potential application in both medical and agricultural settings. Because of it is dual functions (benzylamine and curcumin), with potential dual activity, further in vivo studies and clinical evaluations are necessary for piperidone **3h** to fully explore its therapeutic potential and to address the growing need for new antifungal drugs to combat resistant fungal pathogens.

## Data Availability

The original contributions presented in this study are included in the article/Appendix A. Further inquiries can be directed to the corresponding authors.

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
