# Peer review of "1,5-Diarylidene-4-Piperidones as Promising Antifungal Candidates Against Cryptococcus neoformans"

_antibiotics, 2025, doi:10.3390/antibiotics14090883_

Round 1
Reviewer 1 Report
Comments and Suggestions for Authors
The study is logical and methodical. The experiments are carefully conducted, and the results are well-supported by the data. However, there are significant issues in the writing and clarity that should be addressed before publication. I recommend major revision with attention to the following points:
- While the authors describe their characterization techniques in the Materials and Methods section, the statement in Section 2.1 that compounds were “characterized by ¹H NMR, ¹³C NMR and Mass spectroscopy” is vague. For clarity and completeness, I suggest briefly stating what each technique confirmed—e.g., structural identity, substitution patterns, molecular weights, and purity—so that the reader does not need to refer to the Methods.
- The Introduction section feels abrupt and lacks a smooth narrative flow. Consider elaborating on the limitations of current antifungal therapies (e.g., toxicity, resistance, administration barriers) and clearly stating the rationale for designing this new library of compounds.
- The manuscript contains numerous grammatical errors, tense shifts, and awkward sentence constructions. A thorough language revision is strongly required.
- Reference numbers often appear weird places in sentences. Please revise to integrate references within the body of the sentence or place them after relevant claims.
- Figures 4 through 9 include molecular docking images with amino acid residue labels that are difficult to read. Please revise these figures to improve label clarity and overall readability.
- Provide references for used methodology
- While the docking studies are informative and well-performed, the manuscript would benefit from a discussion of the limitations of in silico predictions. Including a statement about the need for functional validation such as ergosterol quantification or CYP51 enzymatic inhibition assays would strengthen the mechanistic claims.
Author Response
Dear Editor, Dear Reviewers,
We thank the Reviewers for their comments that helped us to improve the manuscript. Please find below our answers to comments. Regards
Dr Marc Maresca
Reviewer 1 :
The study is logical and methodical. The experiments are carefully conducted, and the results are well-supported by the data. However, there are significant issues in the writing and clarity that should be addressed before publication. I recommend major revision with attention to the following points:
1. While the authors describe their characterization techniques in the Materials and Methods section, the statement in Section 2.1 that compounds were “characterized by ¹H NMR, ¹³C NMR and Mass spectroscopy” is vague. For clarity and completeness, I suggest briefly stating what each technique confirmed—e.g., structural identity, substitution patterns, molecular weights, and purity—so that the reader does not need to refer to the Methods.
Answer : All these suggestions have been inserted in the revised version.
2. The Introduction section feels abrupt and lacks a smooth narrative flow. Consider elaborating on the limitations of current antifungal therapies (e.g., toxicity, resistance, administration barriers) and clearly stating the rationale for designing this new library of compounds.
Answer : We thank the Reviewer for this comment. We extend the introduction in the revised version, adding more details on C. neoformans and limitations of current antifungal therapies. We also clearly stated the rationale for the design of the new molecules described here.
3. The manuscript contains numerous grammatical errors, tense shifts, and awkward sentence constructions. A thorough language revision is strongly required.
Answer : We thank the Reviewer for this comment. We performed a thorough language revision.
4. Reference numbers often appear weird places in sentences. Please revise to integrate references within the body of the sentence or place them after relevant claims.
Answer : We thank the Reviewer for this comment. References have been revised.
5. Figures 4 through 9 include molecular docking images with amino acid residue labels that are difficult to read. Please revise these figures to improve label clarity and overall readability.
Answer : We thank the Reviewer for this comment. Figures have been revised.
6. Provide references for used methodology
Answer : We thank the Reviewer for this comment. References have been added in methodology section.
7. While the docking studies are informative and well-performed, the manuscript would benefit from a discussion of the limitations of in silico predictions. Including a statement about the need for functional validation such as ergosterol quantification or CYP51 enzymatic inhibition assays would strengthen the mechanistic claims.
Answer : We thank Reviewer 1 for this valuable comment. We agree that having functional validation of the inhibition using CYP51 enzyme assay and/or ergosterol level measurement will confirm the docking analysis. Unfortunately, we do not have the expertise to perform such assays. Such expertise could be implemented in our lab or through collaborations for continuing this work in a near future. Then, although docking analysis results are strong (as indicated by Reviewer 1), as suggest by Reviewer 1, we clearly stated in the abstract, the limitation of the docking analysis. And at the end of the docking results section of the revised version of the manuscript we added a sentence stating that docking analysis will require confirmation using enzyme assay and/or ergosterol quantification.
Regards
Dr M Maresca
Reviewer 2 Report
Comments and Suggestions for Authors
1] Insert small images of compounds structures next to their NMR spectra.
2] 1H-NMR spectra of 2g missing in SI and 13C-NMR missing compounds 2i, 2j, 2k, 2m
3] There are lot of mislabeled NMR spectra: 1b both 1H & 13C are added twice, please change first one to 1a. 2l 13C-NMR spectra repeating on page 24 (SI)
4] Correct the inverse phase of 2n 13C-NMR on page 26 (SI).
5] Specify the purification conditions of each compound. How they were purified either by crystallization or by column chromatography. If by column chromatography, then specify column dimensions and eluent.
6] The synthesis is not novel and there are 3,5-diarylidene-4-piperidone derivatives that have antifungal activity.
7] Conduct in-vitro enzyme inhibition assays (e.g., CYP51 activity) or cellular studies (e.g., sterol profiling) to validate the docking results. Use a C. neoformans CYP51 structure if available or justify the use of T. cruzi CYP51 with sequence homology data.
8] Shorten the Abstract, rewrite the discussion because it looks like an introduction.
Comments on the Quality of English Language
The discussion is poorly written, and there is no cohesion.
Author Response
Dear Editor, Dear Reviewer,
Thank you for your comments, please find below our answers.
Regards
Dr M Maresca
Reviewer 2 :
1] Insert small images of compounds structures next to their NMR spectra.
Answer: Compound structures have been inserted next to the NMR spectra.
2] 1H-NMR spectra of 2g missing in SI and 13C-NMR missing compounds 2i, 2j, 2k, 2m
Answer : All requested data have been added. when compound was reported in the literature, references have been cited at the right place. To show the purity (>95%) of all compounds, we provided 1HNMR.
3] There are lot of mislabeled NMR spectra: 1b both 1H & 13C are added twice, please change first one to 1a. 2l 13C-NMR spectra repeating on page 24 (SI)
Answer: Thank you. Corrections have been made.
4] Correct the inverse phase of 2n 13C-NMR on page 26 (SI).
Answer: We thank Reviewer 2 for this comment. Corrections have been made; the APT sequence is shown (CH and CH3 down; CH2 and Cq up).
5] Specify the purification conditions of each compound. How they were purified either by crystallization or by column chromatography. If by column chromatography, then specify column dimensions and eluent.
Answer: The details regarding purification conditions (by recrystallization) have been added in the results section.
6] The synthesis is not novel and there are 3,5-diarylidene-4-piperidone derivatives that have antifungal activity.
Answer : We agree that Claisen-Schmidt condensation is known from 1881 and 3,5-diarylidene-4-piperidone derivatives have been reported to have antifungal activity. These reports have been cited in the document (20-22). However, a systematic modification of the substituent pattern revealed that tetrabutoxy groups and the presence of benzyl amine moiety enhanced activity consistently outperforming fluconazole. Based on these SAR findings, butoxy-substituted derivative (3h) was selected for further biological evaluation, with potent activities against broad fungi. In addition, this lead candidate showed a low toxicity.
7] Conduct in-vitro enzyme inhibition assays (e.g., CYP51 activity) or cellular studies (e.g., sterol profiling) to validate the docking results. Use a C. neoformans CYP51 structure if available or justify the use of T. cruzi CYP51 with sequence homology data.
Answer : We thank Reviewer 2 for this valuable comment. We agree that having functional validation of the inhibition using CYP51 enzyme assay and/or ergosterol level measurement will confirm the docking analysis. Unfortunately, we do not have the expertise to perform such assays. Such expertise could be implemented in our lab or through collaborations for continuing this work in a near future. Then, as suggest by another reviewer (Reviewer 1), and since we can perform such assays, we clearly stated at the end of the docking results section of the revised version of the manuscript the limitations of our data and wrote that docking analysis will require confirmation using enzyme assay and/or ergosterol quantification in future works. Regarding the use of CYP51 from T. cruzi rather than the one from C. neoformans, as stated in the manuscript at the beginning of the results section on docking analysis, it is based on the fact that similar compounds described in the literature were docked using T. cruzi enzyme (ref 15 of the initial version of the manuscript Nagargoje, et al. Biodivers. 2020, 17, e1900624, doi:10.1002/cbdv.201900624). In order to compare our data with the ones of this publication, we decided to use the same enzyme (the one of T. cruzi). In addition, as our compounds are active against different fungal species tested and are not exclusively active on C. neoformans, it suggests that all fungal CYP51 can be used, and not only the one from C. neoformans.
8] Shorten the Abstract, rewrite the discussion because it looks like an introduction.
Answer : The abstract and the discussion have been revised as suggested by the Reviewer.
Reviewer 3 Report
Comments and Suggestions for Authors
The manuscript presents an interesting and well-structured study on the synthesis and antifungal evaluation of 1,5-diarylidene-4-piperidones, with potential as novel antifungal candidates against Cryptococcus neoformans and other clinically significant fungi. The integration of chemical synthesis, biological assays, cytotoxicity evaluation, and molecular docking provides a comprehensive approach. However, certain sections require clarification, additional details, and methodological rigor to strengthen the scientific quality. Following points need to be addressed in revised manuscript:
- The abstract is informative but too condensed. Briefly highlight key SAR findings and the rationale for selecting butoxy-substituted derivatives for further study.
- Expand on why neoformans was prioritized over other pathogenic fungi at the initial screening stage.
- The discussion of heteroresistance could include more recent statistics on fluconazole resistance in Cryptococcus spp.
- How the purity of compounds (>98%) was confirmed?
- The MIC assay description is brief; specify whether MIC values were based on visual growth inhibition or optical density.
- State the number of replicates (biological vs. technical) for antifungal tests and cytotoxicity studies.
- Although SAR trends are discussed, consider summarizing them in a concise table or diagram highlighting the effect of substitutions on activity and toxicity.
- Provide a rationale for why 3h (benzyl-substituted) showed significantly better activity compared to others—are hydrophobic interactions alone sufficient?
- Docking results are well-presented but lack validation. Include RMSD values or redocking of fluconazole to confirm docking accuracy.
- Figures 2 and 3 (cytotoxicity) are not very clear.
- Add a legend in Table 2 and Table 3 explaining abbreviations of fungal species for non-specialist readers.
- Strengthen the conclusion by specifying which compound(s) merit progression to in vivo studies and why.
Author Response
Dear Editor, Dear Reviewer
Thank you for your comments that allowed us to improve our manuscript.
Please find below our answers
regards
Dr M Maresca
Reviewer 3 :
The manuscript presents an interesting and well-structured study on the synthesis and antifungal evaluation of 1,5-diarylidene-4-piperidones, with potential as novel antifungal candidates against Cryptococcus neoformans and other clinically significant fungi. The integration of chemical synthesis, biological assays, cytotoxicity evaluation, and molecular docking provides a comprehensive approach. However, certain sections require clarification, additional details, and methodological rigor to strengthen the scientific quality. Following points need to be addressed in revised manuscript:
- The abstract is informative but too condensed. Briefly highlight key SAR findings and the rationale for selecting butoxy-substituted derivatives for further study.
Answer : The abstract has been modified as suggested.
- Expand on why neoformanswas prioritized over other pathogenic fungi at the initial screening stage.
Answer : We thank Reviewer 3 for this comment. Additional information and references have been added to the revised introduction explaining why C. neoformans was prioritized
- The discussion of heteroresistance could include more recent statistics on fluconazole resistance in Cryptococcus
Answer : More information has been added in the introduction of the revised version.
- How the purity of compounds (>98%) was confirmed?
Answer The purity (>95%) was confirmed by 1HNMR.
- The MIC assay description is brief; specify whether MIC values were based on visual growth inhibition or optical density.
Answer : We thank Reviewer 3 for this comment. MIC determination was based both on visual and optical density measurement. It is now clearly stated in the Mat&Meth section of the revised manuscript.
- State the number of replicates (biological vs. technical) for antifungal tests and cytotoxicity studies.
Answer : We thank Reviewer 3 for this comment. However, this information was already indicated in the initial Mat&Meth section of the manuscript.
- Although SAR trends are discussed, consider summarizing them in a concise table or diagram highlighting the effect of substitutions on activity and toxicity.
Answer : A figure (Fig 11) summarizing the MIC and safety index is now provided. Discussion of this figure is also added in page 16.
- Provide a rationale for why 3h (benzyl-substituted) showed significantly better activity compared to others—are hydrophobic interactions alone sufficient?
Answer : We appreciate the reviewer’s request for clarification. In the revised manuscript, we have expanded our discussion of why benzyl-substituted compound 3h (exhibited superior antifungal activity. In addition to potential hydrophobic interactions identified in our docking study, the benzyl group in 3h may enhance antifungal potency through a mechanism similar to that of known benzylamine antifungals such as butenafine. This fact has been discussed in the text and references 26 and 27 have been cited. In 3h, the benzyl substitution may therefore provide a dual advantage.
- Docking results are well-presented but lack validation. Include RMSD values or redocking of fluconazole to confirm docking accuracy.
Answer: We thank Reviewer for this suggestion. MD and RMSD values were calculated and are given in the revised manuscript (Figure 10 and Table 5).
- Figures 2 and 3 (cytotoxicity) are not very clear.
Answer : We thank Reviewer 3 for this comment. Colors were added in the figures 2 and 3 of the revised manuscript.
- Add a legend in Table 2 and Table 3 explaining abbreviations of fungal species for non-specialist readers.
Answer : We thank Reviewer 3 for this comment. Explanations of abbreviations of fungal names have been added into the revised version of the manuscript.
- Strengthen the conclusion by specifying which compound(s) merit progression to in vivo studies and why.
Answer : Thank you. A paragraph has been added in the conclusion specifying that compound 3h is a promising lead candidate.
Regards.
Round 2
Reviewer 1 Report
Comments and Suggestions for Authors
The manuscript is acceptable for publication in its current form
Reviewer 2 Report
Comments and Suggestions for Authors
The revisions provided by the authors successfully resolve all concerns raised during the review process